# Reliability-Aware Multipath Routing of Time-Triggered Traffic in Time-Sensitive Networks

**Kai Huang [1], Xinming Wan [1], Ke Wang [2,\*], Xiaowen Jiang [1], Junjian Chen [3], Qingtang Deng [3], Wenyuan Xu [2], Yonggang Peng [2] and Zhili Liu [4]**

[1] Institute of VLSI Design, Zhejiang University, Hangzhou 310027, China; huangk@vlsi.zju.edu.cn (K.H.); 21831042@zju.edu.cn (X.W.); xiaowen_jiang@zju.edu.cn (X.J.)
[2] Department of Electrical Engineering, Zhejiang University, Hangzhou 310027, China; wyxu@zju.edu.cn (W.X.); pengyg@zju.edu.cn (Y.P.)
[3] Digital Grid Research Institute, CSG, Guangzhou 510670, China; chenjj4@csg.cn (J.C.); dengqt@csg.cn (Q.D.)
[4] Hangzhou Sec-Chip Technology Co., Ltd., Hangzhou 310012, China; liuzhili@sec-chip.com
\* Correspondence: coolwind@zju.edu.cn

**Abstract:** With the development of industrial networks, the demands for strict timing requirements and high reliability in transmission become more essential, which promote the establishment of a Time-Sensitive Network (TSN). TSN is a set of standards with the intention of extending Ethernet for safety-critical and real-time applications. In general, frame replication is used to achieve fault-tolerance, while the increased load has a negative effect on the schedule synthesis phase. It is necessary to consider schedulability and reliability jointly. In this paper, a heuristic-based routing method is proposed to achieve fault tolerance by spatial redundancy for TSNs containing unreliable links. A cost function is presented to evaluate each routing set, and a heuristic algorithm is applied to find the solution with higher schedulability. Compared to the shortest path routing, our method can improve the reliability and the success rate of no-wait scheduling by 5–15% depending on the scale of topology.

**Keywords:** time-sensitive network; routing; fault-tolerant; time-triggered traffic; schedulability

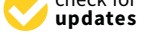



## 1. Introduction

In industrial automation systems, reliable communication with a timing request to deliver messages in Cyber-Physical Systems (CPS) is demanded, which adds constraints in terms of bandwidth and cost [1,2]. Ethernet is extensively used as a solution and has been developed with its advantages of simplicity and flexibility. Data transmission in standard Ethernet is competitive and uncertain due to factors such as queuing. It can only provide best effort (BE) services and cannot guarantee the quality of service (QoS) for time-sensitive streams. Industrial Ethernet (IE) is proposed, such as Ethernet Powerlink, EtherCAT, Profinet and Ethernet/IP, to process time-critical control and motion tasks in production facilities as a result. For these standards, special hardware support and specific integrated circuits are required, which brings limitations to a wide range of applications. IEEE Audio-Video-Bridging (AVB) Task Group has been established to develop the Ethernet-based AVB transmission protocol sets. However, timing is not the only critical aspect to consider, fault tolerance and security are also an essential requirement for CPS [3–5]. Thus, a new set of standards to support safety-critical and real-time applications with zero packet loss and bounded end-to-end latency, namely, Time-Sensitive Networking (TSN), was developed. TSN supports a mixed-criticality communication with three categories of traffic: Time-Triggered (TT) streams, AVB streams and BE streams. TT streams are delivered deterministically and periodical, AVB streams have bounded worst case end-to-end delays (WCD) while BE streams have no timing restriction [6,7].

For deterministic communication, 802.1Qbv protocol is proposed to specify the Time-Aware Shaper (TAS) implementing the time-triggered paradigm at the egress ports of

nodes based on the shared global clock provided by 802.1AS. TAS is essentially a gate mechanism dynamically enabling or disabling the selection of frames from egress queues based on a predefined cyclic schedule called gate control list (GCL) [8]. When the gate is open, a frame can be selected from each queue and transmitted to the physical link in FIFO order, while on the contrary, if the gate is closed, transmission stops. For AVB streams, 802.1Qbv applies a credit-based shaper (CBS) among all queues. A simplified schematic diagram of TAS is demonstrated in Figure 1. In addition to timing requirements, TT streams delivering must be able to tolerate link failure and frame corruption. Therefore, 802.1CB is defined to achieve fault tolerance by frame replication and elimination for reliability (FRER). Redundant copies are transmitted in parallel over multipaths to tolerate any failure of a single link [2]. The number of multipaths are defined as Redundancy Level (RL), which can be specified based on certification standards, such as IEC 61508 for the industrial applications [9].

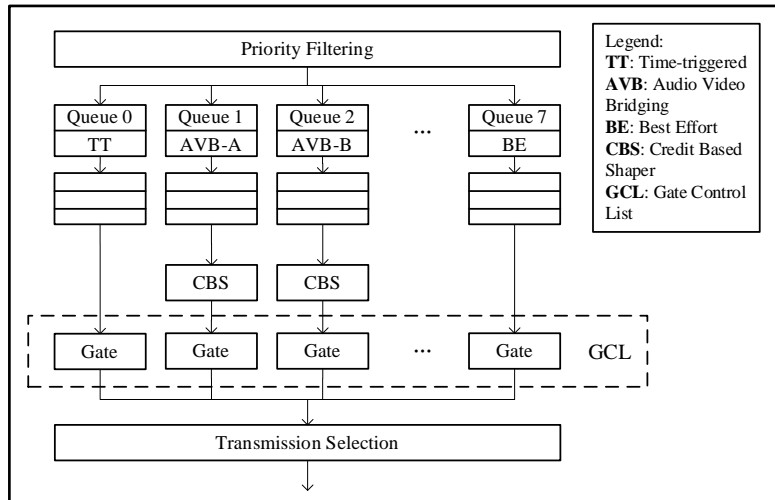

**Figure 1.** Transmission mechanism of Time-Aware Shaper in IEEE 802.1Qbv.

Routing and scheduling of TSN differs greatly in different schedule models and applications due to the fact that these techniques have not been specified in the IEEE 802.1 Standard [10]. Integer linear programming (ILP) and satisfiability modulus theory (SMT) can be used to jointly solve the routing and scheduling problem in [4,11], which minimizes the occurrences where the scheduling algorithm fails to find any available solution. Nonetheless, the aforementioned methods can only handle small instance as the complexity grows rapidly with the scale of the problem [12]. Heuristic algorithms inherently have scalability in solving routing problems [7,13,14]. Regarding the fault-tolerance of TSN routing, many approaches typically replicate the entire network, which is inefficient in terms of cost and power consumption [15]. In order to reduce the cost, redundancy is used to tolerate link failure in [6,9]. However, it is obvious that redundancy will lead to an excessive load in the network, which makes schedule synthesis infeasible. As a result, there is a trade-off between schedulability and reliability. Furthermore, Refs. [6,9,15] consider the reliabilities of all links in the network are consistent. In the industrial application, however, there are cases where the reliability of some links is greatly reduced due to aging, resulting in an inconsistency in transmission qualities. As a result, the reliability of the path containing these links is affected.

In this paper, we present a heuristic-based reliability-aware routing algorithm aiming to enhance the schedulability of time-triggered flows in TSN. We use multipath routing to meet the redundancy level, which is generally ignored in some existing approaches in [4,12,16]. The main contributions are as follows:

- In order to enhance reliability, a novel strategy that selects the candidate routing sets, which extend Shortest Multipath Routing methods to deal with links of different reliability classes is proposed.
- A heuristic algorithm is presented to find the solution with higher schedulability, which is measured by a proposed cost function.
- Test cases based on real industrial application are formulated to prove the effectiveness of the proposed approach.

The remainder of this paper is organized as follows: In Section 2, the related work is discussed. The system model and application scenario are introduced in Section 3. Section 4 presents a motivation example. Section 5 illustrates the proposed Reliability-Aware Multipath Routing in detail. Section 6 discusses the experiment results and evaluation. Finally, the paper is concluded in Section 7.

## 2. Related Work

IEEE 802.1Qbv standards specify network planning as two stages: routing and scheduling. In TSN, typically, routing and scheduling are conducted separate and subsequently. A specific routing scheme such as the shortest path routing determines the data transmission path of each TT stream in the routing stage [17]. On the other hand, the scheduling stage decides the GCL of switches, which is the timing schedule for the opening and closing of the time-controlled gates along the predetermined paths. In the scheduling stage, the algorithm may not find any available schedule for a certain path. Once the number of such unschedulable paths increases, the performance of the network will be greatly affected. To enhance the schedulability, several ILP formulations were introduced for jointly solving the routing and scheduling problem in [4]. Smirnov et al. took the worst-case interference imposed by high-priority traffic into consideration for a mixed-criticality network [16]. The authors of [18] added a pre-processing stage on the basis of [4], introducing a method that pruned unnecessary links and adaptively grouped the subgraph of the network to reduce execution time. However, with the expansion of constraints, the solving time of ILP rises dramatically, making it unsuitable for large-scale problems. Furthermore, it was discussed in [19,20] that the outcome of the routing stage has an impact on the scheduling stage. Recently, approaches were proposed to improve schedulability in the routing phase. Laursen et al. introduced a heuristic to determine the routing of the AVB streams such that their worst-case end-to-end delay is minimized with higher schedulability [7]. Moreover, the load-balance problem was addressed in [14] by distributing the transmission paths among the links as evenly as possible. However, in Reference [7] and [14], the impact of frame size and period on schedulability is not discussed, which lead to a negative effect on the schedule phase. The method presented in [21] used QoS measurement to route TSN flows, which spare the bottleneck links causing the infeasible scheduling.

The aforementioned literature neglect the data corruption and link failure during transmission. Pahlevan et al. introduced a fault injection mechanism to evaluate the reliability of TT communication based on FRER [22], while Smirnov et al. presented another measurement to formally analyze the transmission reliability of switched Ethernet [23]. To achieve fault tolerance, topology and routing were synthesized jointly in [15,24] to obtain architectures that are both fault-tolerant and satisfy the timing request. However, the cost of topology synthesis is huge. In order to reduce the consumption of fault-tolerance, Atallah et al. firstly improved the mean time to failure (MTTF) by temporal redundancy [6] and then used spatial redundancy to meet the RL [9]. Additionally, a schedule mechanism was introduced in that the GCL of TT flows were recalculated first when failure occurred to meet the deadline of the message simultaneously in [25]. Nevertheless, routing methods that take both schedulability and reliability into consideration have not yet been thoroughly investigated.

## 3. System Model

### 3.1. Network Model

A multi-hop switched network is considered according to [7,9], consisting of End Systems (ES) and Switches (B) connected by physical links. The ESs exchange periodic messages according to the static routing table. The network topology is modeled as a directed graph G ($\mathcal{V}$, $\varepsilon$), where $\mathcal{V} = \textbf{ES} \cup \textbf{B}$. The physical connection between vertices $v_i \in \mathcal{V}$ and $v_j \in \mathcal{V}$ is represented as edge $(v_i, v_j) \in \varepsilon$. An example of the topology of the network is shown in Figure 2. The network consists of six ESs and six Bs. All physical links are full-duplex links based on Ethernet. Each edge connecting node $v_i \in \mathcal{V}$ and $v_j \in \mathcal{V}$ is considered to be two separate directed links denoted by ordered pairs $[v_i, v_j]$ and $[v_j, v_i]$, where the first elements are represented as the sender while the second are receivers. The path $p$ is an ordered sequence from the source node $v_s$ to the destination node $v_d$. As shown in Figure 2, one of the paths for stream $s_1$ is $p^i_{(ES0,ES5)} = [ES0, B0, B4, B5, ES5]$. All physical links are full-duplex links based on Ethernet, so each link connecting nodes $v_i$ and $v_j$ is considered to be an ordered pair $[v_i, v_j]$ and $[v_j, v_i]$.

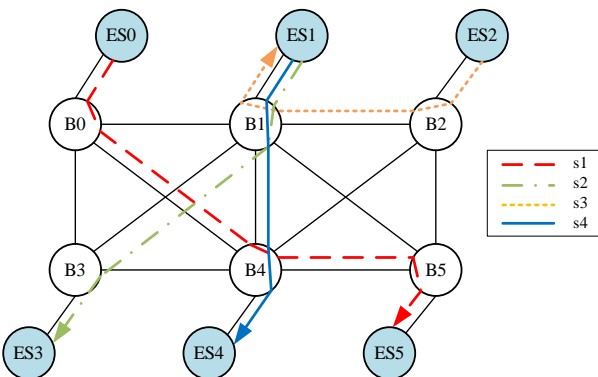

**Figure 2.** Example Time-Sensitive Network (TSN) topology model including 6 End Systems (ESs), which exchange 4 streams via 6 switches.

### 3.2. Application Model

A set of time sensitive applications communicating on TSN is considered on the basis of [6,7,9], which is represented as time-triggered message sets **M**. Each message of **M** is specified by the tuple $m = (v_s, v_d, pr_m, si_m, dl_m, rl_m)$, where $v_s \in \textbf{ES}$ and $v_d \in \textbf{ES}$ are the source node and the destination node. The frame period and size of each message $m$ are defined by $pr_m$ and $si_m$, respectively. The deadline of message $m$ is $dl_m$. Finally, the number of replicas sent by message $m$ is the redundancy level, denoted as $rl_m$. Multiple frames of message $m$ are transmitted in a hyper-period, which is equal to the least common multiple of periods for all messages. A TT stream in TSN contains multiple data frames, unlike TTEthernet, while the latter limits a stream to one data frame in a maximum size of MTU [26]. The set of TT streams in TSN is referred to $S = \{s_m\}$, where $m \in \textbf{M}$. According to the frame replication and elimination for reliability (FRER) protocol in IEEE 802.1 CB, for a stream $s$, each frame in $s$ is replicated at the first bridge and eliminated at the last bridge before the destination [2].

The transmission of data streams is vulnerable to link failure. Normally redundancy is used to solve this problem. We achieve fault tolerance by multipath routing, which generates multiple non-overlapping paths. The definition is given as follows: Non-overlap Routing Set (NRS) is a set of paths between the same source $v_s$ and destination $v_d$ with no shared links. The number of paths in an NRS is equal to $rl_m$. Formally,

$$u_m = \left\{ p^1_m \ldots p^{rl_m}_m \, \middle| \, L^i_m \cap L^j_m = \left\{ \left[ v_s, \, b_{first} \right], \, [b_{last}, \, v_d] \right\} \right\} \tag{1}$$

where $0 < i, j \leq rl_m$. $L_m$ is the set of links contained in path $p_m$, $b_{first}$ and $b_{last}$ are the first and last bridges between $v_s$ and $v_d$, respectively. For an NRS, the path between *ES* and *B* is not counted as an overlapping path because the frame is respective copied and eliminated at the first and last bridge, so there is no overlap on $\left[v_s, b_{first}\right]$ and $[b_{last}, v_d]$.

*3.3. Transmission Reliability*

In industrial applications, cables are usually used for physical connections between devices. Due to the cost constraints, some cables may not be replaced in time after aging. The frequency of failure that occurs on these links is much greater than normal links. Therefore, a scenario is considered: the reliabilities of links connected to different bridges may not be consistent. We divide the links into two categories, namely reliable links and unreliable links.

To measure reliability, there are parameters like Diagnostic Test Interval (DTI) and Mean Time to Detected Error (MTTDE) [6]. However, adopting such parameters makes the case too complicated to analyze. In order to simplify the problem, we use probability to measure the frequency of link failure. The probability model is established based on the model mentioned in [23], which is suitable for switched Ethernets. $P_r$ and $P_{ur}$ are defined as the probability that a failure causing data to be unable to transmit correctly occurs during one communication hop for a reliable link and an unreliable link, respectively. The links connecting ES and B are not considered, for once these links are corrupted, data can never be received completely at the destination node. It should be noted that the values of $P_r$ and $P_{ur}$ do not necessarily need to be accurate, but indicate unreliable degrees, which are determined by the engineer according to the quality of the links in advance. For a certain path $p^i_{[v_s,v_d]}$, the probability of $v_d$ receiving a data frame in a period is as follows:

$$P_{p^i_{[v_s,v_d]}} = (1 - P_r)^j \cdot (1 - P_{ur})^k \tag{2}$$

where $j$ and $k$ are the number of reliable links and unreliable links in the path $p^i_{[v_s,v_d]}$, respectively. As for a certain NRS $u_{[v_s,v_d]}$, the probability of $v_d$ receiving a data frame is defined as:

$$P_{u_{[v_s,v_d]}} = 1 - \prod_{\substack{i \\ 0 < i \leq rl_m}} \left(1 - P_{p^i_{[v_s,v_d]}}\right) \tag{3}$$

where $p^i_{[v_s,v_d]} \in u_{[v_s,v_d]}$.

## 4. Motivation Example

The problem addressed in this paper is defined by the following inputs: (i) a network topology G $(\mathcal{V}, \varepsilon)$, (ii) a set of TT streams *S*. We are interested in determining the multipath routing set $u_m$ of each stream $s_m \in S$ such that they are schedulable with higher reliability.

Take the topology of Figure 2 as an example. Although the throughput of full duplex link is generally 100 Mbps, in order to better visualize the traffic in a hyper-period, we assume the speed is 1 Mbps. The network carries streams $S = [s_1, s_2, s_3, s_4]$ summarized in Table 1 with the topology of **ES** $= [ES0, ES1, ES2, ES3, ES4, ES5]$ and **B** $= [B0, B1, B2, B3, B4, B5]$. We assume edge $(B1, B4)$ and $(B0, B4)$ are unreliable, which are links $[B1, B4]$, $[B4, B1]$, $[B0, B4]$ and $[B4, B0]$. $P_r$ and $P_{ur}$ are 5% and 20%, respectively. If we use the shortest path routing (SPR) approach, we get the result shown in Figure 2. Although this scheme is schedulable, it does not have the ability to tolerant any link failure.

**Table 1.** The *S* used for the motivational example.

| Stream | Source | Destination | Size | Period | Deadline |
|--------|--------|-------------|------|--------|----------|
| $s_1$ | ES0 | ES5 | 10 B | 50 µs | 300 µs |
| $s_2$ | ES1 | ES3 | 20 B | 100 µs | 250 µs |
| $s_3$ | ES2 | ES1 | 20 B | 50 µs | 300 µs |
| $s_4$ | ES1 | ES4 | 40 B | 200 µs | 250 µs |

Consequently, the shortest multipath routing (SMR) can be adopted to achieve fault tolerance, which uses the disjoint route sets [9]. We assume the redundancy level $rl_m = 2$, and the results are demonstrated in Figure 3. However, due to redundancy, the amount of scheduled flows in the network increases, which causes an excessive number of flows to be routed on the shortest path, making the scheme in Figure 3 infeasible. The details of the scheduling results are shown in Section 5.

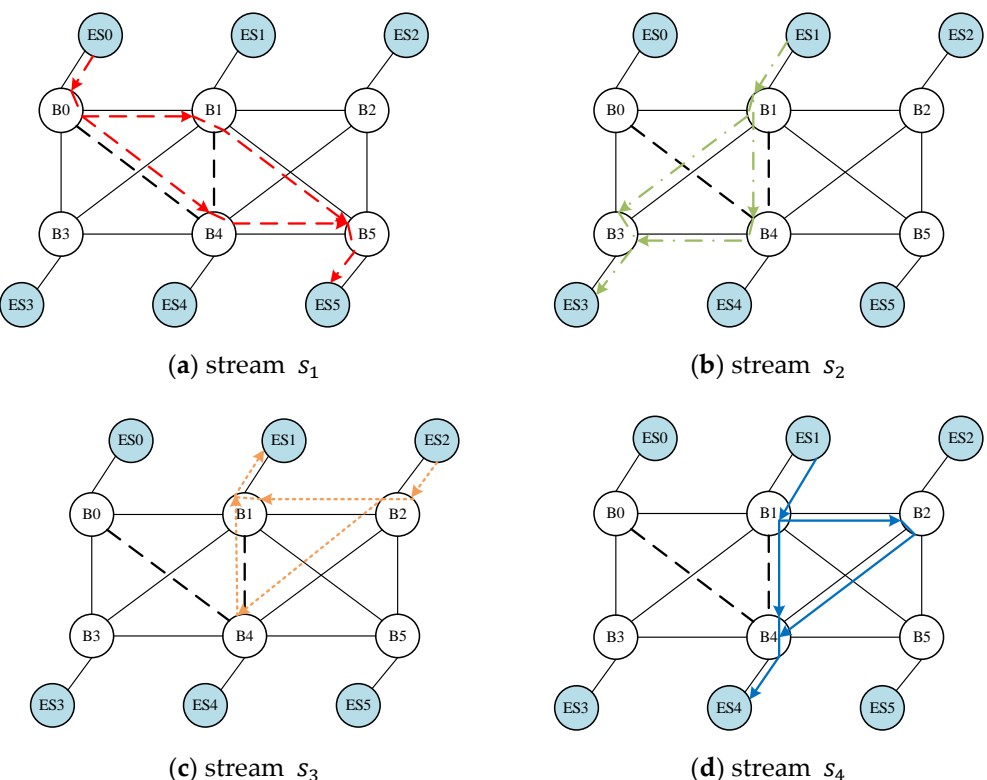

**Figure 3.** Illustrative example of routing results from shortest multipath routing (SMR).

In summary, SPR and SMR approaches have certain limitations. Against these limitations, we present an optimized method to consider both reliability and schedulability. The solution of Reliability-Aware Multipath Routing (RAMR) proposed in this paper is illustrated in Figure 4. Compared with the infeasible scheme in Figure 3, neither the path of $s_1$ nor $s_3$ is the shortest, but the routing scheme in Figure 4 is schedulable. Simultaneously, the reliabilities that are calculated by Equation (3) compared to the schemes in Figures 2 and 3 are shown in Table 2. It can be observed from Table 2 that SMR are more reliable than SPR because of the redundancy, while the reliability of RAMR is higher than that of SMR due to the latter containing some unreliable links.

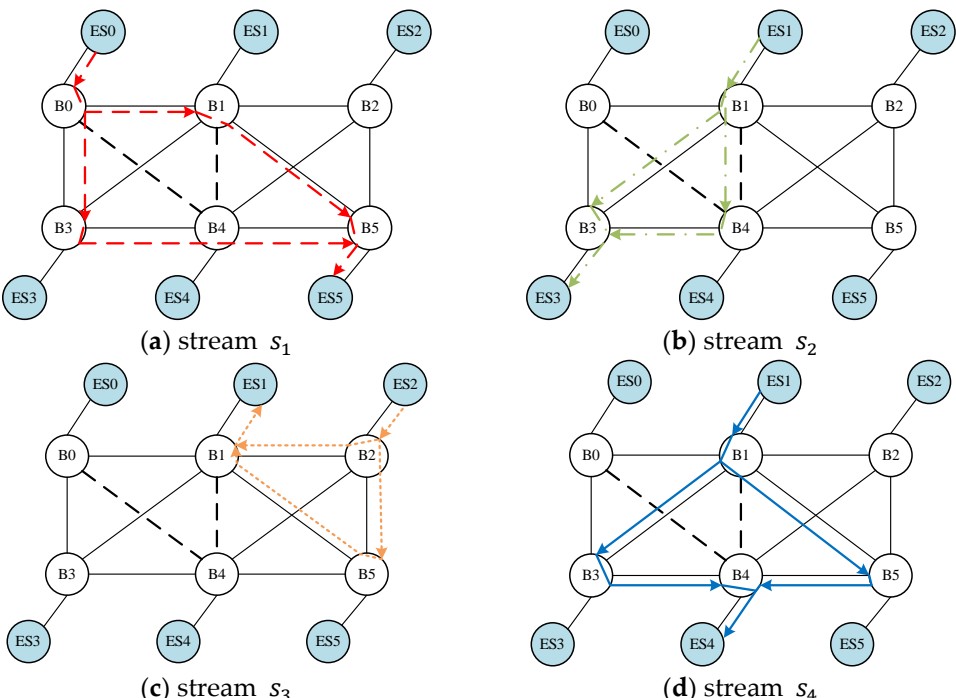

**Figure 4.** Illustrative example of routing results from Reliability-Aware Multipath Routing (RAMR).

**Table 2.** Comparison of the probability of SMR, shortest path routing (SPR) and RAMR routing algorithms.

| Stream | SPR | SMR | RAMR |
|:---:|:---:|:---:|:---:|
| $s_1$ | 76% | 97.7% | 98.6% |
| $s_2$ | 95% | 98.8% | 98.8% |
| $s_3$ | 95% | 98.8% | 99.5% |
| $s_4$ | 80% | 95.2% | 99.1% |
| Average | 86.5% | 97.6% | 99% |

## 5. Reliability-Aware Multipath Routing

Exhaustively enumerating every path between two nodes has been proven to be NP-hard [27]. This requires listing all of the non-overlap path combinations between $v_s$ and $v_d$ of each stream $s_m \in S$, which leads to the need of evaluation on a huge number of combinations. The proposed Reliability-Aware Multipath Routing approach consists of three steps: (i) In the first step, candidate routing solutions that meet the reliability threshold under the scenario discussed in Section 3.3 are found, thereby reducing the search space, as described in Section 5.1. (ii) In the second step, a cost function is presented and used to evaluate each candidate solution, composing of the conflict degree and flow latency. The details are introduced in Section 5.2. (iii) In the third step, we employ the heuristic algorithm of Ant Lion Optimization (ALO) searching the candidate routing solution to minimize the cost function, which is introduced in Section 5.3.

### 5.1. Find Candidate Routing Solutions

As mentioned in Section 4, when several physical links in the topology are unreliable, the probabilities of transmission failure of the non-overlap routing sets, including these links, will rise. Therefore, in order to further improve the reliability of the system, we filter the solutions with a threshold of probability $P_{th}$, presented in Algorithm 1. For each stream $s_m \in S$, the length range of NRSs is computed (line 2). Then we start searching from the shortest NRSs, and remove $u_m^i$ from the set $U_l$ when the calculated $P_m$ does not meet the threshold (line 14–19). After a round of searching, the second shortest NRS is searched,

and so on. When there is no routing set in $U_l$ that meets the threshold, and so is in $U_{l+m}$, where $l + m$ is the second shortest length longer than $l$, the searching is stopped (line 5).

*List* is the output of the algorithm, which is the candidate routing set for each flow. If the *List* of $s_m$ is empty, as there is no routing set that satisfies the probability constraint, either changing the threshold or replacing unreliable links with reliable ones is available to recalculate the outcome. The value of $P_{th}$ has an impact on the reliability of the network. An appropriate threshold can increase the range of candidate routings while ensuring routing reliability, which is beneficial to improve the schedulability of the solution. In scenarios with high security requirements, such as the transmission of a data stream containing critical control information, a higher threshold can be selected. However, the threshold cannot be greater than the probability $P_{u_m}$ where $u_m$ is the shortest NRS that does not contain unreliable links in the network.

In conclusion, Candidate Routing Sets Filtering not only improves the reliability by filtering out routing sets that do not meet the reliability requirement, but also reduces the search space of the heuristic algorithm.

---

**Algorithm 1** Candidate Routing Sets Filtering

---

**Input:** $G(\mathcal{V}, \varepsilon)$, $S$, $P_{th}$
**Output:** *List* for $s_m \in S$
1   **for** $s_m \in S$:
2      calculate the shortest and the longest total length of NRSs $l_{min}$ and $l_{max}$
3      $l \leftarrow l_{min}$
4      $temp \leftarrow 0$
5      **while** $temp \neq 2$ **do**
6        **if** $l > l_{max}$ **then**
7          **break**
8        **else**
9          find all NRSs $u_m^i \in U_l$ with a total length of $l$
10          **if** $U_l = \varnothing$ **then**
11          $l \leftarrow l + 1$
12          **continue**
13        **else**
14          **for** $u_m^i$ in $U_l$ **do**
15            calculate possibility $P_m$ according to (3)
16            **if** $P_m < P_{th}$ **then**
17            remove $u_m^i$ from $U_l$
18            **end if**
19          **end for**
20          **if** $U_l = \varnothing$ **then**
21            $temp \leftarrow temp + 1$
22          **end if**
23          $l \leftarrow l + 1$
23        **end if**
24      **end while**
25      $List \leftarrow \mathrm{U}_{l \in (l_{min}, l_{max})} U_l$
26 **end for**

---

## 5.2. Cost Function

After selecting the list of candidate solutions, a cost function is presented, which is used for the heuristic algorithm discussed in Section 5.3. We define the cost of the solution as the sum of two objectives $O_1$, $O_2$ multiplied by their respective weights $W_1$, $W_2$:

$$Cost(\mathcal{R}) = W_1 \cdot O_1(\mathcal{R}) + W_2 \cdot O_2(\mathcal{R}) \tag{4}$$

The first objective $O_1(\mathcal{R})$ is the sum of the conflict degrees for the routing scheme $\mathcal{R}$. Formally,

$$O_1(\mathcal{R}) = \sum_{i \neq j} \mathcal{D}(s_i, s_j) \tag{5}$$

where $s_i, s_j \in S$. Conflict degree is a measure for the mutual dependence between streams [9,28]. For two streams $s_i$ and $s_j$, the conflict degree $\mathcal{D}(s_i, s_j)$ between them is computed by:

$$\mathcal{D}(s_i, s_j) = (p_i \cap p_j) \cdot \frac{si_i \cdot si_j}{pr_i \cdot pr_j} \tag{6}$$

where $p_i \cap p_j$ is the number of shared links. When there are overlapping paths in $\mathcal{R}$, a larger frame size and smaller period would degrade the schedulability of $\mathcal{R}$. Take the routing scheme of Figure 3 for example. The shared links between $s_2$ and $s_4$ are $[ES1, B1]$ and $[B1, B4]$. The conflict graph $G'(S, D)$ can be drawn based on the conflict degree, where $S$ are the streams routed in the network and $D$ is the set of arc weight equaled to the value of conflict degree between two nodes. Figure 5 is the conflict graph of the illustrative example. The weight of edge $(s_3, s_4)$ is high so that the frame conflict occurs between $s_3$ and $s_4$, resulting in a non-schedulable solution. The no-wait schedule result is shown in Figure 6 as the frames overlap on link $[b_2, b_4]$ between $s_3$ and $s_4$.

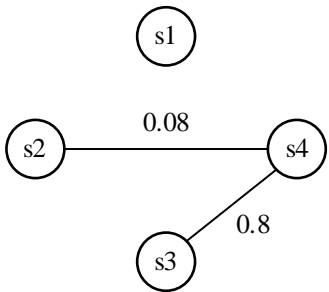

**Figure 5.** The conflict graph of the illustrative example. There are overlapping links between $s_2$ and $s_4$, $s_3$ and $s_4$.

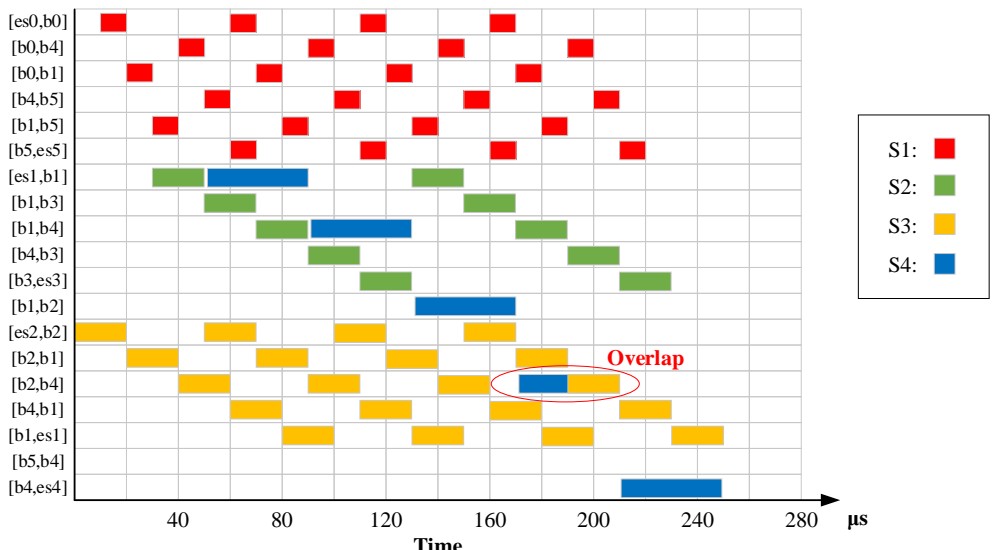

**Figure 6.** The no-wait schedule result of the illustrative SMR scheme in Figure 3.

The second objective $O_2(\mathcal{R})$ reduces flow latency of all streams by maximizing the sum of the difference between the flow latency $\tau(s_m)$ and the deadline $dl_m$ as follows:

$$O_2(\mathcal{R}) = -\sum_{s_m \in S} (dl_m - \tau(s_m)) \tag{7}$$

where $\tau(s_m)$ is defined as the sum of the delays on all links in the routing sets $u_m$. Formally,

$$\tau(s_m) = \sum_{l_i \in u_m} t_{delay}(l_i) \tag{8}$$

Since the heuristic algorithm looks for the solution that minimizes the cost function, $O_2(\mathcal{R})$ is negative. It is foreseeable that a practical implementation will use an individual weight for each flow so that it can be prioritized, but for simplicity, we use a single value weight $W_2$ [7,13].

Our main purpose is to improve the schedulability through the cost function, as a result the weight $W_1$ is much larger than $W_2$, ensuring that the value of the cost function is positive. This way, when the $O_1(\mathcal{R})$ values of different routings are the same, the solution with low flow latency is preferred.

### 5.3. Antlion Optimizer Algorithm

ALO introduced in [29,30] is a meta-heuristic optimization, which searches for a solution that minimizes the cost function in search space. ALO realizes the interaction between ants and antlions through numerical simulation to optimize the problem. The global search is realized by the random walk of ants, and the diversity of population and the optimization performance of the algorithm are guaranteed through the roulette and the elite strategy. Antlion is equivalent to the solution of the optimization problem. It can update and save the approximate optimal solution by hunting ants with high fitness.

The algorithm has three steps as Algorithm 2 demonstrates: (i) First, the candidate routing solution in *List* for $s_m \in S$ from Algorithm 1 is encoded and several initial routing solutions are generated randomly within the feasible region as ants and antlions (line 1). The number of ants and antlions are determined by the size of search space and their respective values of cost function $Cost(\mathcal{R})$ are calculated as fitness (line 2). (ii) Second, the solution with the highest fitness, that is, the solution with the minimal cost function is chosen as the elite antlion in the initial antlion population (line 3). (iii) The antlion for each ant is selected by the roulette strategy and makes the ant walk around the antlion randomly, and finally takes the average value as the position of ant (lines 5–9). The cost function of each ant is calculated and compared to that of the corresponding antlion. When the value of the ant is smaller, it will replace the antlion (lines 10–12). Finally, the antlion with the minimal cost function is regarded as the elite antlion (lines 14–18).

ALO is an iterative algorithm that executes step (ii) and (iii) repeatedly until the maximum iteration time is reached. The fitness of ants and antlions will be recalculated after each iteration and the last elite antlion is the final routing solution. Assuming that the dimension of candidate routings is $n$ and the time to calculate the fitness of a certain routing set is $f(n)$, the time complexity of the ALO algorithm is $O(n + f(n))$. According to the literature [29], the average deviation between the solution obtained by ALO and the best solution for unimodal functions and multimodal functions are in the order of $10^{-3}$ and $10^{-2}$, respectively, while the average deviations of the genetic algorithm are 0.4 and 0.2. ALO benefits from its high convergence speed and local optima avoidance, making it suitable for combinatorial optimization problems.

**Algorithm 2** Antlion Optimizer

---

**Input:** *List*
**Output:** *elite*
1   initialize the first population of ants and antlions from *List* randomly
2   calculate $Cost(\mathcal{R})$ of each ant and antlion
3   *elite* $\leftarrow$ the antlion with the minimize $Cost(\mathcal{R})$
4   **while** the end criterion is not satisfied **do**
5   **for** every ant **do**
6      select an antlion using *Roulette wheel*
7      create *random walk* for ants around the antlion
8      update the position of ant
9   **end for**
10   calculate $Cost(\mathcal{R})$ of all ants
11   **if** $Cost(\mathcal{R}_{ant}) < Cost(\mathcal{R}_{antlion})$ **then**
12      replace the antlion with its corresponding ant
13   **end if**
14   **if** $Cost(\mathcal{R}_{antlion}) < Cost(\mathcal{R}_{elite})$ **then**
15      *elite* $\leftarrow$ *antlion*
16   **end if**
17   **end while**
18   **return** *elite*

---

Figure 7 is the no-wait schedule result of the RAMR approach. Compared to the schedule in Figure 6, The algorithm selects the path where streams $s_1$ and $s_4$ share the link, rather than streams $s_2$ and $s_4$ in Figure 6 due to a smaller conflict degree. Therefore, the routing scheme is schedulable as the schedule has no frame overlap.

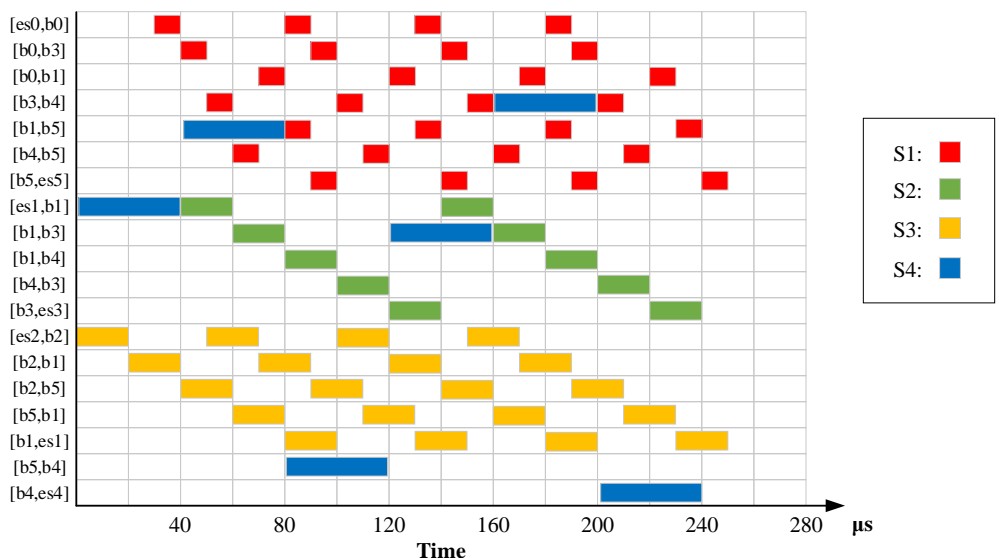

**Figure 7.** The no-wait schedule result of the illustrative RAMR routing scheme in Figure 4.

## 6. Experiments and Discussions

In this section, the performance of the RAMR approach is presented. The setup of the experiment is first discussed. Reliability and schedulability are then evaluated. The results of the experiment are compared to the SPR and SMR methods.

### 6.1. Experiment Setup

We implemented our approach and the comparative SPR and SMR approaches in a Python-based framework. The topologies were created by *NetworkX* [31], which is a package for the creation of complex networks. We used Integer Linear Programming

(ILP) to synthesize schedules by Python API of the Gurobi Optimizer (Ver 9.0.3) [32]. The experiments were run on an Intel Core i5 8300H processor with 2.3 GHz and 8 GB RAM.

Three topologies were used in our test case. The first network in Figure 8 is a small 2D-mesh topology consisting of 6 bridges and 6 end systems. The second network in Figure 9 is a partial mesh topology with the same number of end systems and bridges as Topology 1 but in higher connectivity, which is the same as the illustrated example in the motivation case. The third network in Figure 10 is a topology with the same connectivity compared to Topology 2, consisting of 9 bridges and 6 end systems, which is larger than the first two topologies. The transmission rate is assumed to be 100 Mbps. Flows were generated randomly in the size of $Size = \{100\ \text{B}, 200\ \text{B}, 400\ \text{B}, 800\ \text{B}\}$ and the period of $Period = \{5\ \text{ms}, 10\ \text{ms}, 20\ \text{ms}, 40\ \text{ms}\}$ with redundancy level $rl_m = 2$. $P_r = 5\%$, $P_{ur} = 30\%$ while threshold $P_{th}$ differs according to the scale of topology. The orders of magnitude are based on [18], which simulates the industrial applications. The source and destination of each flow were randomly selected among the end systems. We set the weights of the cost function $W_1$ and $W_2$ as 1000 and 1, respectively. The colony size of ALO is 30, which is approximately equal to the number of flows that need to be scheduled in the network, and the maximum iteration time is 100 as ALO already converged before 100 time iterations. As regards the device characteristics, the delay per bridge was assumed as 4 μs, which is a realistic assumption for state-of-the-art switches [33].

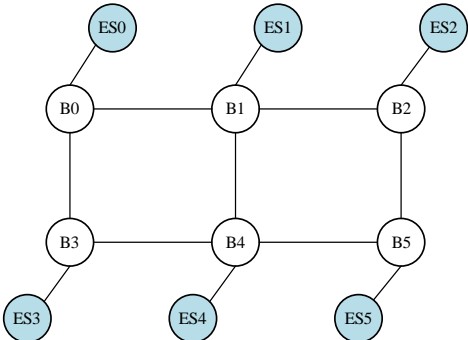

**Figure 8.** Topology 1, which is a small 2D-mesh topology used in our cases.

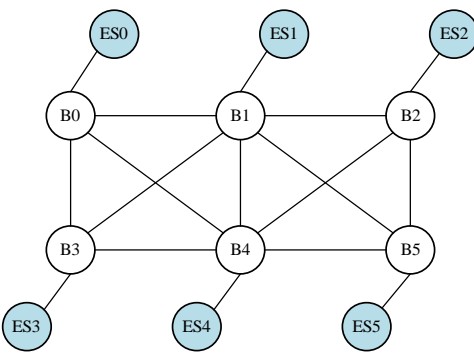

**Figure 9.** Topology 2, which is a partial mesh topology used in our cases.

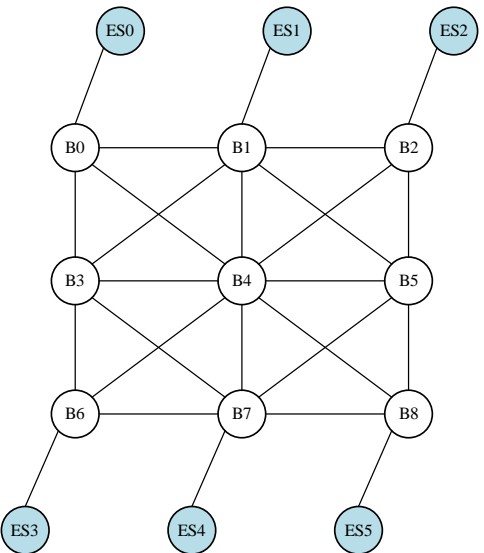

**Figure 10.** Topology 3, which is a larger topology used in our cases.

### 6.2. Reliability

To evaluate the performance of reliability, we gradually increased the number of unreliable links in the topology, observing and comparing the reliability probability based on Equation (3) of our proposed approaches and SMR/SPR approaches. The unreliable links were randomly selected in all edges of the topology. We assume that in Topology 1, $P_{th} = 80\%$ due to a higher threshold will result in an empty set of *List* while $P_{th}$ in Topology 2 and 3 are both 95%. Thirty test cases were formulated in each experimental group.

Figure 11 illustrates that the difference in connectivity will influence the reliability of routings. The vertical axis presents the average value of $P_{u_m}$ from thirty test cases, and the horizon axis presents the number of unreliable links $n$. Regardless of the value of $n$, the reliability of RAMR and SMR is 10% to 20% higher than SPR, which is guaranteed by redundancy.

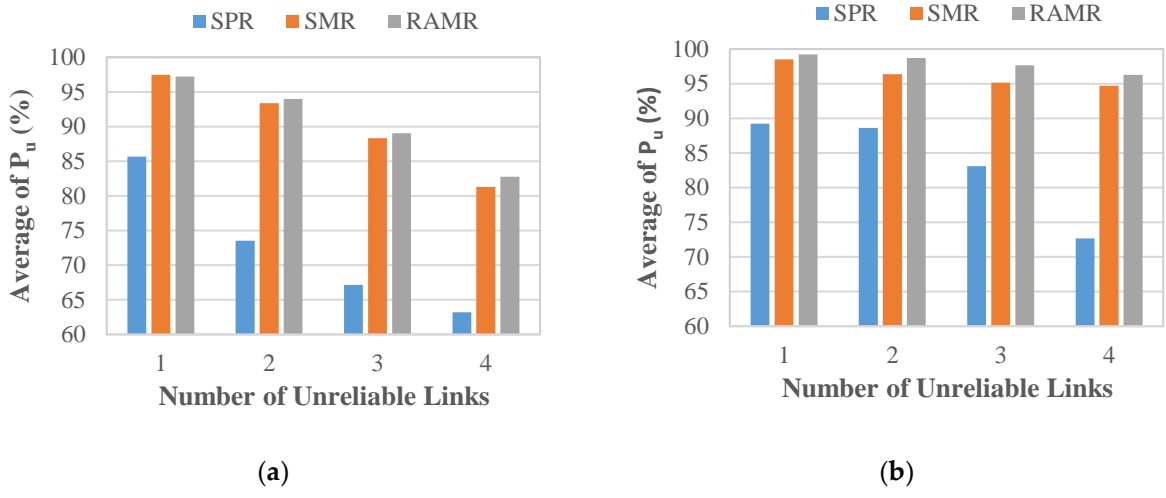

| (**a**) | (**b**) |

**Figure 11.** Reliabilities when the number of unreliable links $n$ varies from 1 to 4 in (**a**) Topology 1; (**b**) Topology 2.

When there are few unreliable links in the network, the difference between SMR and RAMR is not obvious. In Topology 1, as illustrated in Figure 11a, the reliability of RAMR is even slightly worse than that of SMR if $n = 1$. The majority of shortest NRSs in SMR are reliable enough to be added into *List* of each stream as candidate solutions. For a routing scheme $\mathcal{R}$, always selecting the shortest path sets will result in an increase in the

degree of path overlap between different streams, so it is the $Cost(\mathcal{R})$. RAMR prioritizes schedulability, such that the solution is not the shortest. Longer paths correspond to lower reliability, which leads to the results.

Furthermore, for all algorithms, the larger the value of $n$ is, the smaller the reliabilities. This is because the number of unreliable links included in the path sets also increases. However, $P_{u_m}$ of RAMR has a subtler decrease compared to SMR in both two topologies. In Topology 2, when $n$ increases from 1 to 4, the reliability of SMR drops by 16% while RAMR drops by 14% due to some routing sets containing unreliable links that are out of the threshold range. RAMR removes them from *List*, and uses routing sets with higher reliability as candidate solutions. In addition, comparing Topology 1 and Topology 2, the reliability of the former is generally lower than the latter. Topology 2 is a graph with higher connectivity, which has more paths between two certain nodes, providing RAMR with a larger range of options. Therefore, RAMR in Topology 2 improves reliability more significantly than that of Topology 1.

Moreover, we also investigate how the scale of topology will affect the reliability performance. We respectively employ three methods on Topology 2 and Topology 3, in which the number of end systems is the same while bridges differ. In Topology 2, due to the excessive number of unreliable links, the threshold value cannot be maintained, $P_{th}$ changes from 95% to 85% when $n \geq 6$. The experimental results are illustrated in Figure 12. It can be observed that the reliabilities of larger topology in all approaches are better than small topology when $n$ is consistent. In particular, in Topology 3, RAMR has the most significant improvement in reliability compared to SMR when $n = 8$, reaching 6%. Comparing Figure 12a,b at the same time, we can drop the conclusion that RAMR improves reliability more obviously in larger topologies. Since the larger scale of topology contains more bridges, the amount of links that can replace unreliable links increases, which enhances the transmission quality of routing paths.

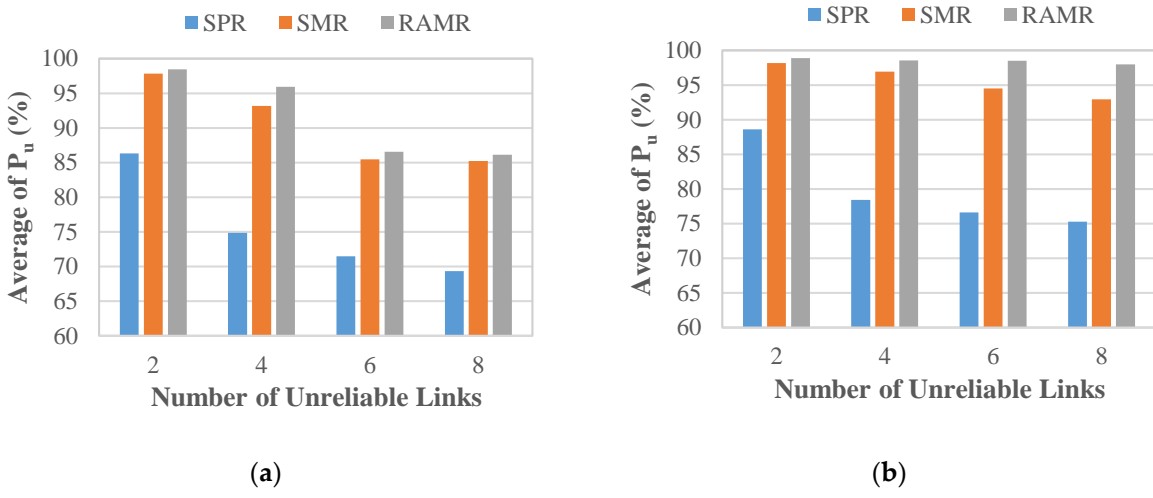

**Figure 12.** Reliabilities in different topologies when the number of unreliable links *n* varies from 2 to 8. (**a**) Reliabilities in Topology 2; (**b**) Reliabilities in Topology 3.

In summary, our method does not significantly improve reliability in low-connective graphs and small topologies. Even to ensure schedulability, there is a chance that it is worse than the shortest path algorithm with redundancy. However, when the scale of topology increases and the connectivity becomes higher, the advantages of the RAMR method gradually manifests.

### 6.3. Schedulability

To evaluate schedulability, 50 random patterns were conducted in each experiment to obtain the success rate of schedule synthesis. It is noted that the schedule algorithm

we used is based on Reference [34] with additional no-wait constraints introduced in [35], which compress the schedules to ensure low latency. We make a comparison between SMR and RAMR methods for they are both based on redundancy. The results with various stream loads are presented in Figure 13. It can be observed that 10-streams cases are almost schedulable in all topologies, yet with the number of stream increases, the success rate of SMR is dropped sharply because the SMR algorithm makes routing selection focus on some of the shortest paths, resulting in a higher conflict degree. On the contrary, the proposed method prefers the routing paths that are more likely to be scattered in the network or which are not prone to frame overlap. Consequently, the success rate of RAMR is visibly higher compared to SMR by 10% on average. In particular, since the conflict between paths increases with the number of flows, the advantages of the proposed routing method enhance at the larger network with more flows, up to 16% in experiments.

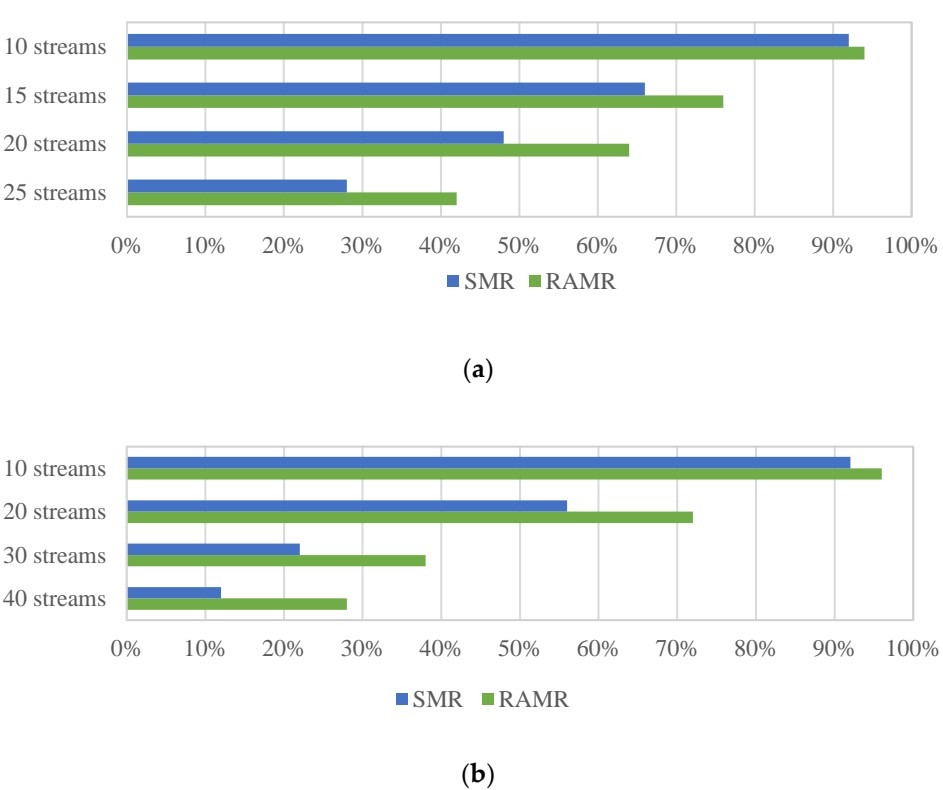

**Figure 13.** The average success rate of schedule among 50 test cases for varying load in: (**a**) Topology 2; (**b**) Topology 3.

Moreover, the trade-off between reliability threshold $P_{th}$ and schedulability was investigated. In the experiment, we assume that there are two unreliable links. Fifteen streams were routed on the network of Topology 2 with different values of $P_{th}$. We formulated 100 test cases each time to get the success rate, as shown in Figure 14. The threshold represents the reliability requirements of the network, which is the lower limit, and the value influences the size of the search space. When $P_{th} = 99\%$, the routing sets in *List* are without any unreliable link, resulting in a small range of alternative paths. Consequently, the space in which the heuristic algorithm can optimize the cost is also limited so that the schedulability decreases. With the drop of the threshold, the search space expands and success rate increases. However, when the threshold is reduced to 97%, almost all paths in the network meet the reliability requirements. Even if $P_{th}$ drops to 90%, the search space will not change much, and the trend of schedulability growth tends toward saturated.

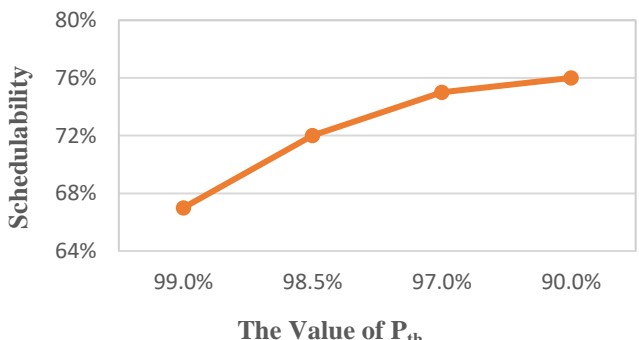

**Figure 14.** Schedulability trend with threshold changes in Topology 2.

## 7. Conclusions

In this paper, a novel method is introduced for the routing of Time-Triggered streams in TSN, which considers a scenario in which the network contains unreliable links. The proposed RAMR approach determines the routing to meet the reliability threshold for each stream and enhance the schedulability by the ALO metaheuristic used to optimize the cost function. Three experiments are formulated to demonstrate the performance of RAMR in different scale topologies with various numbers of flows loaded. The results demonstrate that compared to conventional routing methods, the schedulability has significant enhancement with a higher reliability in handling larger-scale networks. In our future work, we are going to extend the scheduling approaches used in our experiment to improve the schedulability further, while also taking worst-case end-to-end delay (WCD) into consideration in the routing stage, thereby the flow latency brought by redundancy can be reduced.

**Author Contributions:** Conceptualization, X.W. and K.W.; methodology, K.H., X.W. and K.W.; software, X.W.; validation, K.H., X.W. and X.J.; formal analysis, X.W., K.W. and X.J.; investigation, X.W., J.C., and Q.D.; resources, K.H., X.J., W.X. and Y.P.; data curation, X.W., W.X., Y.P., and Z.L.; writing—original draft preparation, X.W.; writing—review and editing, K.H., X.W., K.W. and X.J.; visualization, K.H.; supervision, K.H. and X.J.; project administration and funding acquisition, K.H. All authors have read and agreed to the published version of the manuscript.

**Funding:** This work was supported by the National Key R&D Program of China (2020YFB0906000, 2020YFB0906001).

**Data Availability Statement:** The data presented in this study are available on request from the corresponding author. The data are not publicly available due to privacy.

**Conflicts of Interest:** The authors declare no conflict of interest.

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
