# Peer review of "Reliability-Aware Multipath Routing of Time-Triggered Traffic in Time-Sensitive Networks"

_electronics, doi:10.3390/electronics10020125_

Round 1

Reviewer 1 Report

The authors proposed heuristic-based routing method to achieve fault tolerance by spatial redundancy for TSNs containing unreliable links. The paper is well motivated and the methodology seems to be sound. There are a few comments for the authors to consider. 1. How close is the heuristic algorithm to the optimal solution? 2. What is complexity of the heuristic algorithms? 3. It looks that there are three topologies in the experiments. I suggest the authors present the topologies more clearly to avoid confusion.

Reviewer 2 Report

This paper presents an approach to improve the handling of redundancy
in time-sensitive networks (TSN) to increase the reliability of network
traffic in case of link failures. While the state of the art in TSN is
to use Shortest Multi-Path Routing (SMR) to tolerate link faults, the
authors extend this approach by assuming links have either one of two
reliability levels.
This way, they can model different types of communication links in their
optimisation for the the minimal routing paths. The authors call this
new approach "Reliability-Aware Multipath Routing" (RAMR).

The technical approach is well presented with a good motivation and definition
of the application model.

Language-wise the paper has numerous minor grammar/spelling mistakes.
It would be advised to have an English native to proof-read the paper.

The related work would be more readable if a style is used to name the authors,
e.g., "1stSurname et al. wrote ... [ref]."

p7: It would be useful to talk more explicit about how to chose the P_th value.
(or its problems to chose a generally valid value)

p2/main contributions:
Say much more explicit what your technical contributions are:
RAMR: Extending SMR to deal with links of different reliability class

Additional comments:
p1/15: real-time application --> real-time applications
28: Ethernet are extensive --> Ethernet is extensive
35: is established --> has been established
37: also the essential --> also an essential
p2/51: to tolerant link --> to tolerate link
71: contained these links are affected. -->
containing these links is affected.
p3/111: formal --> formally
p5/169: To be noted --> It should be noted
p6/194/Fig.3: Put the stream id directly to the caption of the individual sub-figure,
which simplifies the reading. (do the same also for other figures where it applied)
p8/232: Try to put Alg.1 on the same page (using a float environment)
p10/Alg.2: The 'for sm in S' is at the wrong place
p15/392: optimize cost --> optimize the cost
